# Improving accuracy of GPT-3/4 results on biomedical data using a retrieval-augmented language model

**David Soong[1], Sriram Sridhar[1]\*, Han Si[1], Jan-Samuel Wagner[2], Ana Caroline Costa Sá[1], Christina Y. Yu[1], Kubra Karagoz[1], Meijian Guan[1], Sanyam Kumar[3], Hisham Hamadeh[2], Brandon W. Higgs[1]**

**1** Translational Data Sciences, Genmab, Princeton, New Jersey, United States of America, **2** Data Sciences and AI, Genmab, Princeton, New Jersey, United States of America, **3** Commercial Data Sciences, Genmab, Princeton, New Jersey, United States of America

☯ These authors contributed equally to this work.

\* srsr@genmab.com

**Data Availability Statement:** Supplemental data, including 19 questions used to evaluate LLM performance, responses from different LLMs, and reviewer scoring, are included in the accompanying

## Abstract

Large language models (LLMs) have made a significant impact on the fields of general artificial intelligence. General purpose LLMs exhibit strong logic and reasoning skills and general world knowledge but can sometimes generate misleading results when prompted on specific subject areas. LLMs trained with domain-specific knowledge can reduce the generation of misleading information (i.e. hallucinations) and enhance the precision of LLMs in specialized contexts. Training new LLMs on specific corpora however can be resource intensive. Here we explored the use of a retrieval-augmented generation (RAG) model which we tested on literature specific to a biomedical research area. OpenAI's GPT-3.5, GPT-4, Microsoft's Prometheus, and a custom RAG model were used to answer 19 questions pertaining to diffuse large B-cell lymphoma (DLBCL) disease biology and treatment. Eight independent reviewers assessed LLM responses based on accuracy, relevance, and readability, rating responses on a 3-point scale for each category. These scores were then used to compare LLM performance. The performance of the LLMs varied across scoring categories. On accuracy and relevance, the RAG model outperformed other models with higher scores on average and the most top scores across questions. GPT-4 was more comparable to the RAG model on relevance versus accuracy. By the same measures, GPT-4 and GPT-3.5 had the highest scores for readability of answers when compared to the other LLMs. GPT-4 and 3.5 also had more answers with hallucinations than the other LLMs, due to non-existent references and inaccurate responses to clinical questions. Our findings suggest that an oncology research-focused RAG model may outperform general-purpose LLMs in accuracy and relevance when answering subject-related questions. This framework can be tailored to Q&A in other subject areas. Further research will help understand the impact of LLM architectures, RAG methodologies, and prompting techniques in answering questions across different subject areas.

supplemental information file. The code base for the retrieval-augmented LLM will be available at https://github.com/genmab upon publication of this manuscript.

**Funding:** The author(s) received no specific funding for this work.

**Competing interests:** The authors have declared that no competing interests exist.

## Author summary

Large language models (LLMs) have recently made a significant impact on the field of general artificial intelligence and are beginning to be incorporated for a variety of tasks across industries. Their utility in generating precise information pertaining to specific subject areas is actively being explored. Here we presented application of a retrieval-augmented generation (RAG) LLM, which utilizes literature specific to cancer research, and compared its performance to three other general purpose LLMs (e.g. GPT-4) in answering questions specific to cancer research. We found that the RAG model produced generally more accurate and relevant answers to questions about treatment and biology of a specific blood cancer, while general purpose LLMs GPT-4 and GPT-3.5 had generally more readable answers but with more instances of incorrect information (i.e. hallucinations). This work showcases a practical application of LLMs in cancer research and attempts to evaluate how augmenting LLMs with credible source information can help improve their utility in a research setting.

## Introduction

The development of large language models (LLMs), such as bidirectional encoder representations from transformer (BERT) and generative pre-trained transformer (GPT) models, has revolutionized the field of natural language processing [1–4]. Applications of these LLMs have ranged from sentiment analysis and machine translation to code generation and question answering across different areas of focus (domains) [5–9]–all demonstrating remarkable performance. However, despite their impressive execution and widespread use, LLMs are designed for general purpose use and often lack domain-specific knowledge and vocabulary. They can also perpetuate biases based on skewed content in the data they were trained on, and need to be further refined through reinforcement learning and alignment approaches to understand user intentions while making them more truthful and less toxic [10,11]. Furthermore, concerns have been raised about the potential for LLMs to generate misleading information, or hallucinations, which can have severe implications in areas such as scientific research and healthcare, among others. These issues can severely hamper the utility of these models, as was seen with Meta's Galactica [12,13]. In biomedical and clinical research, accuracy and reproducibility of research methods and results and appropriate sourcing of information are key tenants. The generation of false or misleading information in response to scientific or clinical questions presents a roadblock in the broader utility and adoption of these models in a research or clinical setting and would need to be addressed for further application of LLMs in these areas.

 Popular LLMs with billions of parameters such as GPT-3 [14], PaLM [15], OPT [16], and LLaMA [17] are typically trained on vast amounts of information collected from the Internet (e.g. the Common Crawl dataset [18]) and capture a diverse range of language patterns and knowledge. In the simplest terms, LLMs are deep neural networks that use prior knowledge from training data to predict the next set of words constituting a response. They are built on principles of the transformer architecture [4] first introduced by Google in 2017. LLMs break down words or word fragments into units referred to as tokens. They use models called token embeddings as a way to convert tokens to a numerical vector representation that captures semantic meanings in a high-dimensional space. Additionally, positional embeddings are created to contain positional information about each input token. These token and positional embeddings are combined and fed into a portion of the architecture called an encoder, which

understands the context of the embeddings based on data it has been trained on. The decoder portion of the architecture then generates an output based on what has been processed in the encoder. Within the encoder and decoder, the attention mechanism allows the model to understand how tokens are related to one another. Depending on the implementation, some models use the encoder-decoder architecture (e.g. T5 [19]) while most others use only the decoder (e.g. the GPT family models). Furthermore, the encoder architecture can be used to generate text embeddings (hereafter simply referred to as embeddings) to numerically represent the concepts of full sentences, paragraphs, or documents for applications such as semantic search, text clustering, topic modeling, and classification [20].

As the size and complexity of language usage in a corpus increases [21–23], LLMs can learn to capture the diversity of language usage across different domains and genres and generate natural language responses when prompted on scientific literature, social media, or news articles, with equal ease [24]. With increasing size, a wide-ranging corpus can inadvertently incorporate a significant amount of noise or irrelevant data, resulting in a reduced signal-to-noise ratio [25]. This may adversely affect the generated text's quality, leading to decreased coherence, meaning, or accuracy. As LLMs use transformer architectures to understand context and generate responses, if the broad corpus has inherent biases which affect how information is contextualized, the responses will subsequently reflect these biases as well. A corpus that predominantly features one type of language or cultural context may display bias towards that specific domain or culture [26].

One approach to addressing these limitations is to retrain or finetune an LLM with a focused corpus tailored to a specific domain [25,27], thereby reducing the risk of generating irrelevant or misleading information and enhancing the reliability and precision of the LLM's outputs in specialized contexts. Numerous publications have highlighted the efficacy of domain specific LLMs in their respective fields. For example, BioBERT [28] targets biomedical text mining tasks, SciBERT [29] and PubMedBERT [30] address scientific literature, and LEGAL-BERT [23] specializes in legal text processing. However, retraining LLMs to encompass new documents might be impractical due to the cumulative computational and data scientist resources required per update. The LLM architecture might also need to be updated to incorporate more parameters to memorize more facts [31]. As LLMs have demonstrated extraordinary abilities to learn in-context information purely from its prompt [14], retrieval-augmented generation (a.k.a. RAG) approaches have proven promising [27,32]. RAG models utilize a domain-specific corpus created along with the lexical indices or embeddings. When a user enters a prompt or query, these models retrieve relevant context from the corpus using lexical search (e.g. BM25 [33]) or a pretrained/fine-tuned semantic retriever (e.g. Spider [34], OpenAI embeddings [20]), and then seed a pre-trained LLM with such context to then generate a response to the query. The LLM here is only being used to understand context and generate a response, therefore avoiding the prohibitive time and cost of retraining the LLM. The use of domain-specific corpora allows RAG models to source from both relevant and recent information. In the context of scientific research, the ability to use recent and specific content (e.g. articles from peer reviewed journals) provides a source of "ground truth" from which the model can extract information, while avoiding the computational and financial costs associated with continuous retraining of models [35].

In this study, several LLMs were evaluated to investigate if a RAG approach on a focused corpus could improve the accuracy of an LLM's applications in biomedical Q&A. Three scoring metrics were utilized to compare outputs between models using a set of evaluation-based questions focused on disease characterization, genetic subtypes, treatment options, and clinical outcomes in diffuse large B-cell lymphoma (DLBCL). This exercise demonstrates a practical use case of applying LLMs in facilitating scientific research, and highlights the pros and cons of LLMs based on the information they are drawing from.

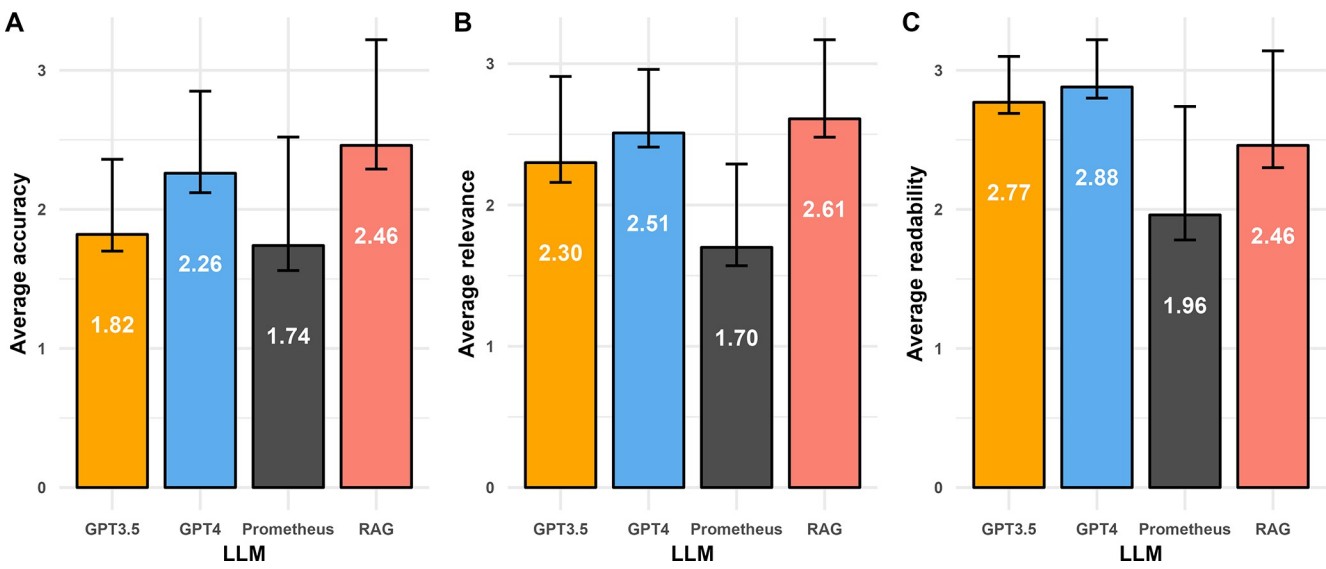

**Fig 1.** Average (A) accuracy, (B) relevance, and (C) readability scores across 19 questions/queries answered, per LLM. Each bar represents the average score across 19 questions for the three metrics (error bars represent standard error of the mean).

## Results

The performance of the LLMs varied across the questions tested and metrics assessed. In terms of accuracy, the RAG model built using GPT-3 on DLBCL publications outperformed the other LLMs tested. Across the 19 questions tested, the RAG model had the highest average score (2.46), followed by GPT-4 (2.26), GPT-3.5 (1.82), and Prometheus (1.74) (**Fig 1A**). When looking at the number of questions with high accuracy scores for each model (**Table 1**), the RAG model had the highest proportion of questions scoring above 2.5 on average (12/19). GPT-4 had the next highest proportion (7/19), followed by Prometheus (4/19) and GPT-3.5 (2/19). When accounting for each individual reviewer score on accuracy across all questions asked (57 total scores from three reviewers across 19 questions), the RAG model again had the highest proportion of 3-point scores (36/57) compared to the other models, which did not yield more than 22 3-point scores (**Fig 2A**). Conversely, Prometheus had the lowest average accuracy across questions and the highest number of 1-point scores across all questions and reviewers (29/57). Prometheus and GPT-3.5 had the lowest number of questions scoring above 2.5 (4 and 2, respectively).

Interestingly, Microsoft's Prometheus was the only model to not score a value of 1 on accuracy for question #6 ("What is the overall response rate of DLBCL patients treated with glofitamab?"). Numerical overall and complete response rates (ORR and CRR, respectively) reported by GPT-3.5 (ORR = 65.1%, CRR = 35.1%) and GPT-4 (ORR = 62.7%, CRR = 39.2%) were not

**Table 1. Number of questions scoring at least 2.5 or more per metric (Accuracy, Relevance, Readability).**

| | Number of questions averaging score above 2.5 | | |
|---|---|---|---|
| LLM | Count (Accuracy) | Count (Relevance) | Count (Readability) |
| **GPT-4** | 7 | 10 | 17 |
| **GPT-3.5** | 2 | 7 | 16 |
| **Prometheus** | 4 | 2 | 6 |
| **RAG** | 12 | 14 | 11 |

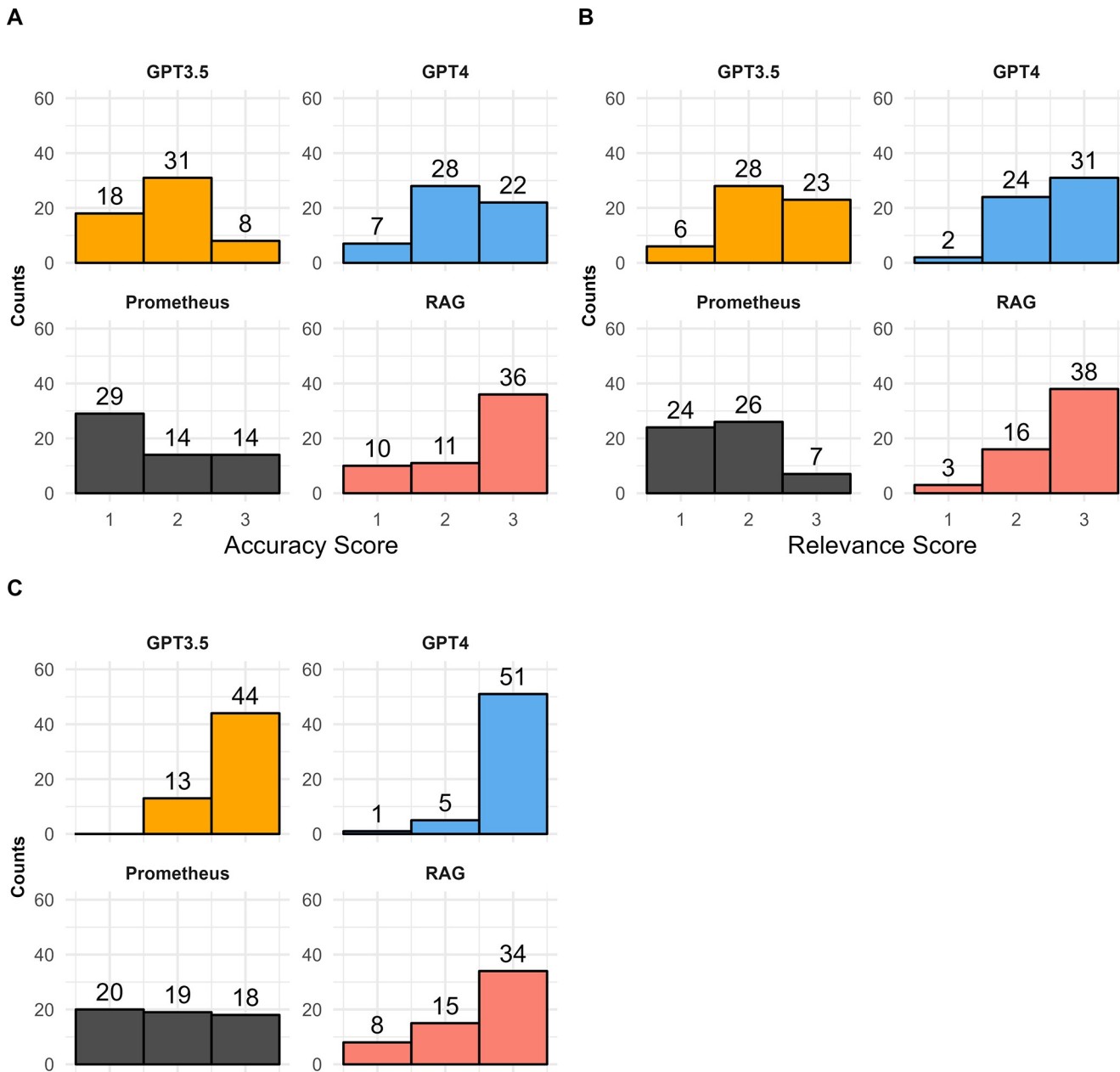

**Fig 2.** Histogram of (A) accuracy, (B) relevance, and (C) readability scores for each LLM. Bars represent the count of 3-point, 2-point, and 1-point scores from each of the three reviewers for each of the 19 questions (57 scores in total per LLM).

consistent with their references cited and had either fabricated or provided incorrect references. Microsoft's Prometheus model scored a value of 2 because there was a mixture of accurate and inaccurate answers to the question, i.e., this model accurately captured the ORR value of glofitamab treatment (52%) in the Dickinson et al, NEJM reference [36], but also incorrectly used the median duration of objective response rather than median duration of CR. The RAG model result was not accurate in answering this question because the official glofitamab trial

efficacy paper [36] was not available on PubMed Central (https://www.ncbi.nlm.nih.gov/pmc/) and therefore not included in the corpus.

In terms of relevance, the RAG model performed slightly better than GPT-4 and GPT-3.5. The RAG model had the highest average relevance score (**Fig 1B**) across the 19 questions (2.61) which was slightly higher than the average relevance scores from GPT-4 (2.51) and GPT-3.5 (2.30), with Prometheus performing worst on average (1.70). The RAG model again had the most questions scoring above 2.5 (**Table 1**) on average for relevance (14/19) compared to GPT-4 (10/19), GPT-3.5 (7/19) and Prometheus (2/19). When looking at all individual reviewer scores for relevance across all questions (**Fig 2B**), the RAG model had the highest proportion of 3-point scores (38/57), with GPT-4 having slightly fewer (31/57). Both the RAG model and GPT-4 had the lowest proportion of 1-point scores (3/57 and 2/57, respectively), suggesting that the attempted answers from these two LLMs was generally inferring the proper context. Prometheus conversely had the highest proportion of 1-point scores (24/57) suggesting that the LLM may have in several instances misconstrued what the question was asking in generating its answers.

The irrelevant answers (i.e. low scoring questions) across all LLMs were primarily due to references to other diseases or treatment. For example, in question #14 ("Have checkpoint inhibitor treatments in monotherapy or combination therapy settings shown efficacy in DLBCL patients? Provide references."), the GPT-4 model cited three references, one of which was in Hodgkin's lymphoma (DLBCL is a non-Hodgkin's lymphoma) and another that discussed CAR-T, which is not a checkpoint inhibiting drug agent, though the model associated this treatment modality with immunotherapies and extended relevance to CAR-T therapies. GPT-3.5 also cited a reference evaluating a checkpoint inhibitor treatment in Hodgkin's lymphoma.

When assessing readability, GPT-4 and GPT-3.5 performed better than the other two LLMs. On average, readability scores (**Fig 2C**) were highest for GPT-4 (2.88) and GPT-3.5 (2.77) when compared to Prometheus (1.96) or the RAG model (2.46). GPT-4 and GPT-3.5 also had the highest proportion of questions with high readability scores (17/19 and 16/19), respectively (**Table 1**) compared to the RAG model (11/19) and Prometheus (6/19). Across all questions and reviewers, the proportion of 3-point readability scores were also highest for GPT-4 (51/57) and GPT-3.5 (44/57), with the other two models not having more than forty 3-point readability scores (**Fig 2C**). Readability was particularly low scoring in the clinical category of questions for the RAG model (**S1C Fig**), compared to accuracy (**S1A Fig**) and relevance (**S1B Fig**) scores in the same question category. Microsoft's Prometheus model once again scored last in this category (18/57 3-point scores), primarily due to concise, yet vague answers, often with little detail. For example, for question #7 ("What is a treatment to use in DLBCL patients who have progressed on CAR-T?"), the Prometheus model simply reported references without summarization, including one study where multiple drugs were approved, and referenced only those of approved agents, ignoring studies evaluating investigational drug agents. The readability of the answers produced by these LLMs, GPT-4 and GPT-3.5 in particular, speaks to the "generative" abilities of these models. GPT-4 was (at the time of writing) the most advanced of the four models in terms of number of parameters and trained data, and the readability of GPT-4 answers in this exercise is consistent with this idea.

Across the 19 questions, both the GPT-3.5 and GPT-4 LLMs generated a higher number of answers with at least one hallucination (**Table 2**) (13/19 and 8/19, respectively) compared to the RAG model and Prometheus (3/19 and 4/19, respectively). These hallucinations were primarily associated with fabrication of both references and clinical results. Although LLMs are known to be behind in mathematical capabilities [37], the inaccuracy of numerical results appeared to be due to hallucinations or context understanding rather than limitations in mathematical reasoning.

**Table 2. Count of questions with at least one hallucination in answer across LLMs.**

| LLM | No hallucinations in answer | Answer contained at least 1 hallucination |
|---|---|---|
| GPT-4 | 11 | 8 |
| GPT-3.5 | 3 | 13 |
| Prometheus | 15 | 4 |
| RAG | 16 | 3 |

These results suggest that the performance of LLMs can vary widely depending on the specific task and domain. Overall, the RAG model enhanced with domain specific data outperformed other LLMs with respect to accuracy and relevance of answers, and produced answers with the fewest observed hallucinations. The newest of the four general purpose LLMs (GPT-4) outperformed the others in terms of readability of answers. It should be noted that this evaluation was limited to a specific set of questions and metrics using a one-shot (i.e. one prompt and answer, no follow-up prompting), and further research is needed to fully understand the strengths and limitations of different LLMs.

## Discussion

The benefits and drawbacks of using LLMs trained on broad corpora versus a RAG approach ultimately are dependent on the specific use case and desired outcomes. In biomedical and healthcare research, it is important to have accurate, relevant, and unbiased information supported by published literature to address clinical and scientific questions. In this study, quantifying the accuracy and utility of LLMs was conducted for answering qualitative and quantitative biomedical questions related to the treatment and prognosis of patients with DLBCL. Results here demonstrated that the RAG model performed better on biomedical-specific tasks than the other LLMs evaluated, specifically with respect to accuracy and relevance of results. This suggests that RAG models can provide more accurate and reliable information for specific fields, reducing the likelihood of generating irrelevant or misleading outputs, while maintaining the flexibility and adaptability of a general purpose LLM.

One major advantage of the RAG model is the easy integration of new domain knowledge that the base LLM may not have been trained on. When a new document is added to the corpus, the model only needs to calculate the embeddings to facilitate retrieval during future queries. On the other hand, fine-tuning or retraining an LLM on a new corpus takes both time and resources (computational and developer). Since the RAG model needs to prompt a pre-trained LLM into performing specific tasks such as summarizing across relevant documents and extracting information without using prior knowledge, the model typically uses a large amount of tokens as input and multiple iterations of base LLM inference (i.e. text-completion API) calls, which can increase the compute cost in its application. The dependence on a certain LLM (e.g. OpenAI GPT-3) also implies that the desired prompt behavior needs to be closely monitored when the LLM backend is updated with new training data, or when the user switches to a different base LLM (e.g. GPT-4, Dolly 2 [38], Open Assistant [21], or RedPajama [39]).

The performance of the RAG is bound by the limitations of the base LLM's vocabulary (tokenizer) and internal representation of concepts (embeddings). For example, question #13 asked about minimal residual disease (MRD) in DLBCL, but the document retriever returned articles about MRD in multiple myeloma and chronic lymphocytic leukemia—two distinct hematological malignancies from DLBCL. The RAG model here relied on GPT-3 as the summarization engine which failed to distinguish between the different disease types, leading to an incorrect answer. These issues may be ameliorated by utilizing more sophisticated document

retrieval methods. For biomedical literature, domain specific models such as BioBERT and PubMedBERT can be used for tokenization and embedding calculation; additional metadata filters can also be used to improve relevance of retrieved documents. As an example, when the retrieval method was modified in the RAG model to directly search on PubMed for supporting articles by significance, the model provided informative and relevant answers detailing the measurement of disease clones with V(D)J sequences, as well as the association with clinical outcomes.

Overall, general LLMs provide highly readable and coherent text when answering questions on various subjects. The performance of the RAG model demonstrated the utility of pre-trained LLMs as a backend in performing various reasoning tasks through purposely crafted prompts. Prompt engineering has been an active area of research that continues to expand the capability of pre-trained LLMs through methods such as: zero-shot [40], few-shot [14], chain of thought [41], tree of thoughts, self-ask [42], and ReAct [43] reasoning. These reasoning properties allow LLMs to be used as programmable agents to orchestrate and perform tasks across different domains (e.g. ToolFormer [44], Visual ChatGPT [45], Langchain [22], GPT plugins [46]). Prompting approaches that decompose complex questions into smaller sub-questions might help the RAG retrieve more relevant documents while maintaining specificity. However, for simplicity of evaluation, this study only assessed performance of zero-shot prompting (where only the answer to the initial prompt was evaluated).

Though findings from this study were informative, it had several limitations that need to be considered. The assessment included only 19 questions which accounted for various clinical, therapeutic, and biological content, attempting to address pertinent context in biomedical research. The question set while relevant was certainly not exhaustive. The focus was on a single disease (DLBCL), which may or may not be generalizable to other diseases or domains. In addition, the scoring metrics selected included accuracy, readability, and relevance, which might not have captured other important aspects of the text such as strength, completeness, and consistency. For simplicity, the scoring was performed across the entire answer as opposed to by sentence or phrase within an answer, which might have provided more granularity on the LLMs performance. While scoring questions in this manner can be subjective, we adjusted for this by using a set of multiple reviewers with varying degrees of research expertise to provide 3 independent reviews of each question, accounting for variance among reviewers. Questions were also specific enough such that available literature could be used to assess accuracy of answers. A point of emphasis for the evaluation of responses was to look for factually incorrect answers (hallucinations), which were more likely to garner the lowest scores, as opposed to answers which were factually correct but not exhaustive. Factual inaccuracy ranged from incorrect statements to links or mentions of references which did not exist. The RAG model included an arbitrary number of full-text articles (1,868), which might not have represented the most relevant or comprehensive set of articles for the disease. It is possible that an optima of accuracy, relevance, and readability can be achieved with a RAG model by increasing the size and breadth of the corpus, and future work will be needed to test this hypothesis. This is somewhat mitigated by the fact that the RAG model is only using papers from the corpus that have the highest similarity score to the prompt, so as long as papers relevant enough to answer the question are in the corpus, increasing the corpus size may only provide marginal improvement. On the other hand, in a rapidly evolving field such as biomedical research, a larger corpus with more recent research might present new theories or findings that contradict previous beliefs. A RAG model with more sophisticated reasoning and information retrieval mechanisms might be required to review relevant articles in the context of factors such as recency, impact, quality of research, and source credibility.

Another consideration when building the biomedical corpus for the RAG model and other LLMs is proper representation of all demographic groups–particularly those from low- and middle-income groups. A question such as "Is there a difference in biomarker testing for DLBCL among Caucasian, Asian, and African American populations?" relies on the training corpus used by these LLMs and the RAG model to be comprehensive enough to have information available to answer this question. However, it is entirely possible that comprehensive studies on biomarker testing performed in DLBCL (or any other disease) do not adequately account for variations in gender, ethnicity, or socioeconomic status of DLBCL patients overall. As a result, any answer given by any LLM would either be deficient or incomplete in addressing this question. As LLMs become more pervasive and widely used, these AI models need to address such biases in their training data for equitable application and effectiveness of the models in global health contexts.

As the field of AI evolves at a rapid pace, the ability to apply these approaches to newer generations of these models needs to be constantly explored. Rapid advancement and development of foundation models across text, image, video, and other data modalities necessitates the adaptation of AI in a fair, accurate, and reliable fashion to maximize impact on healthcare and drug development. Integration of generative AI into everyday clinical practices still faces hurdles such as safety and effectiveness, liability, and data privacy [47,48], underscoring the need for ongoing research to address these challenges and fully harness AI's transformative power in medicine. Despite the considerations mentioned, this study provides valuable insights into the performance of LLMs on different types of corpora and highlights the importance of domain-specific knowledge in enhancing the accuracy and relevance of LLMs in their application. This work also provides a practical example of how LLMs can be used to facilitate and streamline common tasks associated with biomedical research.

## Methods

### Evaluation framework

The performance of generically trained LLMs was tested versus a retrieval-augmented generation (RAG) model in question answering (Q&A) tasks related to disease biology and drug development. A set of 19 questions/queries (**Table 1**) focused on mechanisms and treatments associated with diffuse large B-cell lymphoma (DLBCL) were provided to evaluate LLM performance. The questions covered a broad range of topics related to DLBCL disease biology including clinical and molecular subtypes, genetic subsets and relevant biomarkers, clinical management, and standards of care and other available therapies. Questions were designed to look for both qualitative and quantitative answers (e.g. overall response rate and prevalence of genomic alterations). The questions were not chosen based on any pre-defined paradigms, but rather to allow for generation of general and specific information on topics pertaining to DLBCL biology and drug development. Each question was provided to four different LLMs: Open AI's general ChatGPT-3.5 [49], OpenAI's general GPT-4 [49], Microsoft's Prometheus model (based on GPT-4 [50]), and a RAG model (based on GPT-3) using a custom set of full-text publications associated with DLBCL. The queries intentionally varied in detail to assess the ability of each LLM to infer the expected result. For example, question #15 provided a concise query for DLBCL diagnosis and prognosis, while question #3 asked specific treatments for a target in the disease with accompanying references to support the answer.

The two general GPT-based LLMs from OpenAI were only trained on content up to September 2021 (OpenAI GPT-4 Technical Report [51]), as opposed to Microsoft's Prometheus and the RAG models. Release versions of GPT-4 and GPT-3.5 used to answer the questions

were from 3/23/23 to 4/28/23 (updates were released on a weekly or bi-weekly basis and were documented).

## RAG model and dataset

Scientific papers were downloaded from PubMed Central (PMC [52]) using the Entrez E-utilities [53]). Each of the following search terms was used to retrieve up to 500 articles (per term): 'diffuse large b-cell lymphoma', 'follicular lymphoma', 'epcoritamab', 'glofitamab', 'minimal residual disease', 'ctDNA'. By default, Entrez returns articles sorted by PMC identifier. The search terms used were meant to generate a corpus specific to DLBCL, related biomarkers, standards of care, and therapeutic options, not to specifically answer the questions used in this evaluation. This created a unique dataset of 1,868 full-text articles, which constituted the corpus used by the RAG model. The documents were first pre-processed to exclude potentially unstructured or noisy text (e.g. figures, tables, references, author disclosure) and split into segments of 4,000 tokens. Embeddings were then calculated using the OpenAI model text-embedding-ada-002 and stored in a local database. For each of the 19 questions in **Table 3**, the query was transformed into an embedding vector and compared to the database of embeddings

**Table 3. Questions used for LLM evaluation classified into group and scope categories.**

| Question # | Question | Group | Scope |
|---|---|---|---|
| 1 | What is epcoritamab? Please provide sources for your answer. | Drug information | General |
| 2 | What are the subtypes of DLBCL? Please provide sources for your answer. | Disease biology | General |
| 3 | What are the antibody therapies targeting CD20 for treatment of DLBCL? Please provide sources for your answer. | Drug information | General |
| 4 | What is the standard of care for treatment of DLBCL? | Clinical information | Specific |
| 5 | What are the approved drugs for treatment of DLBCL? | Clinical information | Specific |
| 6 | What is the overall response rate of DLBCL patients treated with glofitamab? | Clinical information | Specific |
| 7 | What is a treatment to use in DLBCL patients who have progressed on CAR-T? | Drug information | General |
| 8 | What are common treatments used in patients who have relapsed or were refractory to standard of care treatments in DLBCL? | Drug information | General |
| 9 | Do any DLBCL patient subtypes respond more favorably to chemotherapy or CAR-T treatments? | Clinical information | Specific |
| 10 | What are the most common adverse events observed in DLBCL patients treated with R-CHOP? | Clinical information | Specific |
| 11 | What biomarkers in DLBCL have been reported to correlate with either response or progression following treatment with R-CHOP? | Clinical information | Specific |
| 12 | What treatment combinations have been shown to be effective in DLBCL patients who have progressed on CAR-T treatment? Please provide sources for your answer. | Clinical information | Specific |
| 13 | How can minimal residual disease (MRD) be used to understand clinical outcomes in DLBCL patients? Please provide sources for your answer. | Disease biology | General |
| 14 | Have checkpoint inhibitor treatments in monotherapy or combination therapy settings shown efficacy in DLBCL patients? Provide references. | Drug information | Specific |
| 15 | DLBCL diagnosis and prognosis. | Clinical information | General |
| 16 | Landscape of DLBCL treatment as SOC. Please provide sources for your answer. | Clinical information | Specific |
| 17 | Emerging novel treatment options for DLBCL patients. | Drug information | General |
| 18 | what is the importance of TP53 in DLBCL? | Disease biology | General |
| 19 | What is the prevalence of double hit mutations in lymphoma? | Disease biology | Specific |

**Table 4. Prompts for GPT3 in the retrieval-augmented workflow.**

| Stage | Prompt |
|---|---|
| Stage one | Instruction: You are a truthful AI assistant. You answer questions only based on provided context below. If the context is not relevant to the question, say you do not know the answer. No need to explain why.<br>Context: {segment of article}<br>Question: {user query}<br>Answer: |
| Stage two | Please combine the following paper's summaries. Only use the context below and not incorporate any prior knowledge.<br>Paper #1: {answer 1 based on segment 1}<br>Paper #2: {answer 2 based on segment 2} |

(generated from the 1868 article corpus) using cosine similarity. The top $k$ document segments by similarity were retrieved and formed the knowledge context for the user query. The synthesis of the answer to the query was achieved in two stages: in stage one, text-davinci-003 was used to answer the query using each of the $k$ context segments with prompt instructions to minimize inclusion of non-factual information from the LLM. This generated $k$ answers which were combined into a final response in the second stage using another call to text-davinci-003 with a summarization prompt (**Tables 4–5**). Various $k$ values were explored; in this paper we presented results from $k$ = 5 to balance between corpus coverage and token length limitation.

## Evaluation metrics

Answers were scored for each question on a three-point scale (1–3, with 3 being highest) based on three metrics: accuracy, relevance, and readability (**Table 6**) by eight independent reviewers. Each reviewer was assigned to a subset of questions such that the metrics for the 19 questions were scored 3 times each, providing an average and variance for each metric and question (e.g. the RAG model's answer to question #3 could have an average accuracy score of 2.67 and variance of 0.33, based on reviewer scores of 3, 3, and 2). Answers to all questions could be found via search. Accuracy and relevance assessments focused on factual correctness of answers, correctness of references or links to references, or general pieces of knowledge included or not included in an answer. The 3-point scale used for each evaluation category also allowed for some granularity in scoring answers. For example, an answer might be given a score of "2" if the result was factually correct but links to supporting references were broken or incorrect. An answer which did not directly address the question being asked or contained factually incorrect information (i.e. hallucinations) might garner a score of "1" for accuracy. As both the language model and oncology therapeutics fields are constantly evolving, there is some recency bias associated with answers to questions and the data which LLMs are trained on. This was in part accounted for through the types of questions chosen and the scale used to assess responses. An emphasis of the evaluation was to specifically look for factually incorrect answers, as opposed to incomplete answers which may be a result of recency bias. Reviewers

**Table 5. Workflow and LLM descriptions used in this study.**

| Workflow | Evaluation | Base LLM |
|---|---|---|
| RAG model | Python workflow | text-davinci-003 |
| chatGPT3 | OpenAI web | chatGPT3 (gpt-3.5-turbo) |
| chatGPT4 | OpenAI web | chatGPT4 (gpt-4) |
| **Prometheus** | Microsoft web | Custom GPT4 |

**Table 6. Answer scoring metric descriptions for LLM comparison.**

| Metrics | Score | | |
|---|---|---|---|
| | 1 | 2 | 3 |
| Accuracy | Mostly inaccurate or misleading content | A mix of accurate and inaccurate content | Factually accurate and reliable content |
| Relevance | Mostly irrelevant content | Partially relevant content | Highly relevant and on-point content |
| Readability | Difficult to read, unclear or convoluted language | Moderately readable, with some unclear passages | Easy to read, clear, and concise language |

were all Ph.D. level scientists with an average of 8 years of biopharma industry experience and 11 years of post-doctoral work experience. The prompts were stratified into three high level categories based on relevance to drug information, disease biology, and clinical information. Questions were also grouped based on being general (i.e. high level) or specific (i.e. asking for details) to assess performance between LLMs using questions answers requiring different levels of detail. Complete information on questions, answers, and scoring of each question across the four LLMs tested is provided in the Supplementary Data (**S1 File**) for this manuscript.

## Supporting information

**S1 Fig.** Boxplot of average score per question for each LLM model. Each point represents the average (A) accuracy, (B) relevance, and (C) readability score for a single question (out of 19 total). Points are colored by the question category.
(TIFF)

**S1 File. Detailed breakdown of 19 questions, provided answers from LLM, three reviewer scores for accuracy, relevance, and readability per question, notes from reviewers (where relevant) explaining rationale for provided score, reviewer name, and annotation (1 = yes, 0 = no) for whether a hallucination was observed with an answer.** This set of information is provided for each of the for LLMs tested, one per worksheet. Also provided in a separate worksheet is the question grouping used to categorize questions in **S1 Fig**. The last worksheet contains details of answers provided by the RAG model when varying the number of answers (k) the model used to generate a final answer. Additional information on the papers used to generate answers and intermediate answers the model used to generate the final output are also given.
(XLSX)

## Acknowledgments

The authors thank Bryan Ho and Swathi Vangala for their contributions to infrastructure.

## Author Contributions

**Conceptualization:** Sriram Sridhar, Han Si, Jan-Samuel Wagner, Brandon W. Higgs.

**Data curation:** David Soong, Ana Caroline Costa Sá, Christina Y. Yu, Kubra Karagoz, Meijian Guan.

**Formal analysis:** David Soong, Sriram Sridhar, Han Si, Ana Caroline Costa Sá, Christina Y. Yu, Kubra Karagoz, Meijian Guan, Sanyam Kumar.

**Methodology:** David Soong, Sriram Sridhar.

**Supervision:** Sriram Sridhar.

**Validation:** Sanyam Kumar.

**Visualization:** David Soong, Sriram Sridhar, Han Si.

**Writing – original draft:** David Soong, Sriram Sridhar, Han Si, Jan-Samuel Wagner, Hisham Hamadeh, Brandon W. Higgs.

**Writing – review & editing:** David Soong, Sriram Sridhar, Han Si, Jan-Samuel Wagner, Hisham Hamadeh, Brandon W. Higgs.

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
