## [Decision Letter · Decision Letter 0]

24 Jan 2024

PDIG-D-23-00346

Improving accuracy of GPT-3/4 results on biomedical data using a retrieval-augmented language model

PLOS Digital Health

Dear Dr. Sridhar,

Thank you for submitting your manuscript to PLOS Digital Health. After careful consideration, we feel that it has merit but does not fully meet PLOS Digital Health's publication criteria as it currently stands. Therefore, we invite you to submit a revised version of the manuscript that addresses the points raised during the review process.

Please submit your revised manuscript within 60 days Mar 24 2024 11:59PM. If you will need more time than this to complete your revisions, please reply to this message or contact the journal office at digitalhealth@plos.org. Please include the following items when submitting your revised manuscript:

We look forward to receiving your revised manuscript.

Kind regards,

Raymond Francis Sarmiento, MD

Section Editor

PLOS Digital Health

Journal Requirements:

1. Please provide separate figure files in .tif or .eps format only and remove any figures embedded in your manuscript file. Please also ensure that all files are under our size limit of 10MB.

Additional Editor Comments (if provided):

Reviewers' comments:

Reviewer's Responses to Questions

**Comments to the Author**

1. Does this manuscript meet PLOS Digital Health’s publication criteria? Is the manuscript technically sound, and do the data support the conclusions? The manuscript must describe methodologically and ethically rigorous research with conclusions that are appropriately drawn based on the data presented.

Reviewer #1: Yes

Reviewer #2: Yes

2. Has the statistical analysis been performed appropriately and rigorously?

Reviewer #1: Yes

Reviewer #2: No

3. Have the authors made all data underlying the findings in their manuscript fully available (please refer to the Data Availability Statement at the start of the manuscript PDF file)?

Reviewer #1: Yes

Reviewer #2: No

4. Is the manuscript presented in an intelligible fashion and written in standard English?

Reviewer #1: Yes

Reviewer #2: Yes

5. Review Comments to the Author

Reviewer #1: In this paper, the authors aim to reduce to practice the use of LLMs as a knowledge agent in the pharmaceutical/biotech setting by evaluating various LLMs, including a retrieval augmented one, in their ability to answer several canonical questions about DLBCL. Expert reviewers scored the LLM outputs in terms of accuracy, relevance, and readability and find that the retrieval augmented model performs best on aggregated, although uniformly. This work represents an early case study in the utility of LLMs in the biotech/pharmaceutical setting and so its significance is highly time dependent, meaning with each passing quarter year models will improve, domain-specific models will evolve, and key challenges will be addressed, e.g. alignment and model updating.

The findings reported herein are useful for practioners and the main novelty lies in the demonstration that a retrieval augmented model can outperform others.

The authors constructed a good case study in that the 19 questions posed represent a good cross-section of typical questions spanning biological, general, and clinical and the use of 8 expert reviewers with a central assessor provides sufficient redundancy to estimate ‘ground truth’ scores.

As is noted by the authors, the study does have limitations – the number of questions assessed is relatively speaking small (N=19); the focus on a sole disease area begs the question of how well these findings would generalize; the better performance may be driven by choices made in the scoring matrix, e.g. relative over-weighting of penalizing hallucinations over incomplete answers. 

In addition to the limitations noted by the authors, the paper raises other questions: what is the variance in the expert reviewers’s scoring and how does this relate to the variance in model performance for specific questions? For the retrieval augmented model, how does performance correlate with parameters, e.g. k used in document retrieval or the literature corpus supplied? 

The paper would benefit from these additional analyses.

In summary, the authors present a case study of general interest to biotech/pharmaceutical audiences who are keen to reduce to practice LLMs for knowledge base purposes and provide an implementation of retrieval augmented models that could be used widely. The paper is clearly written, the findings substantiated, and the methodology is written such that others practiced in the art could reproduce it.

I support publication with revision.

Reviewer #2: In general, the idea is clear and presents a solution to a common problem in LLMs such as hallucinations. However, the following comments stand out:

Minor comments:

- In the introduction, on line 60, the concept of embeddings is introduced, however the introduction is abrupt and without context. I would suggest introducing the concept of embeddings in introduction, where the concept of retrieval-augmented is introduced so that the reader can have a context of what embeddings are and their use in this specific use case.

- Figure 2 can be confusing, I would suggest restructuring it in another way (e.g. bar plots) or if necessary move it to supplementary.

- It would be important to know more details about the experiment so that it can be replicated as best as possible. Values such as the temperature of the models, or the value of k used to choose the top k document segments.

- Although this is a method that improves the performance of existing language models, it should be mentioned in the discussion that the use of generative language models is still far from being a reality in such clinical practice.

Major comments:

- One of the greatest weaknesses of generative language models is the poor capacity for reproducibility and determinism in their responses. For this reason, it is suggested to run the experiments more times and capture the differences in responses. Likewise, a more rigorous statistical analysis of the results is suggested, presenting confidence intervals and statistical tests.

- Although mentioned in the discussion, the possible biases and limitations in the approach should be expanded. The use of scientific articles from PubMed as a data source may not represent all populations, mainly populations less represented in scientific research, such as populations from low- and middle-income countries (LMICs), and therefore, although the approach reduces some limitations of LLMs such as the retraining needs, we must also be very careful as it could introduce other biases. Likewise, some of the questions must represent this type of problems so that rare cases are also evaluated.

6. PLOS authors have the option to publish the peer review history of their article (what does this mean?). If published, this will include your full peer review and any attached files.

**Do you want your identity to be public for this peer review?** For information about this choice, including consent withdrawal, please see our Privacy Policy.

Reviewer #1: No

Reviewer #2: Yes: David Restrepo

---

## [Decision Letter · Decision Letter 1]

2 Jul 2024

Improving accuracy of GPT-3/4 results on biomedical data using a retrieval-augmented language model

PDIG-D-23-00346R1

Dear Mr. Sridhar,

We are pleased to inform you that your manuscript 'Improving accuracy of GPT-3/4 results on biomedical data using a retrieval-augmented language model' has been provisionally accepted for publication in PLOS Digital Health.

Best regards,

Raymond Francis Sarmiento, MD

Section Editor

PLOS Digital Health

Reviewer Comments (if any, and for reference):

Reviewer's Responses to Questions

**Comments to the Author**

1. If the authors have adequately addressed your comments raised in a previous round of review and you feel that this manuscript is now acceptable for publication, you may indicate that here to bypass the “Comments to the Author” section, enter your conflict of interest statement in the “Confidential to Editor” section, and submit your "Accept" recommendation.

Reviewer #1: All comments have been addressed

Reviewer #2: All comments have been addressed

2. Does this manuscript meet PLOS Digital Health’s publication criteria? Is the manuscript technically sound, and do the data support the conclusions? The manuscript must describe methodologically and ethically rigorous research with conclusions that are appropriately drawn based on the data presented.

Reviewer #1: Yes

Reviewer #2: Yes

3. Has the statistical analysis been performed appropriately and rigorously?

Reviewer #1: Yes

Reviewer #2: Yes

4. Have the authors made all data underlying the findings in their manuscript fully available (please refer to the Data Availability Statement at the start of the manuscript PDF file)?

Reviewer #1: Yes

Reviewer #2: Yes

5. Is the manuscript presented in an intelligible fashion and written in standard English?

Reviewer #1: Yes

Reviewer #2: Yes

6. Review Comments to the Author

Reviewer #1: The authors have addressed the concerns I laid out in my review namely providing more details on the model performance vis a vis variance etc.

Reviewer #2: The authors have clearly worked on improving evaluations through the inclusion of more human evaluators, and the reliability of the results by dealing with model variability by adding response variance. Furthermore, although the article has some limitations (as in any research project), the authors recognize them.

I think that the use of RAGs is a method that presents very great potential for cases in which you do not have access to the models, computational resources, or data for the generation or fine-tuning of a Large Language Model. It seems to me that the article is relevant.

7. PLOS authors have the option to publish the peer review history of their article (what does this mean?). If published, this will include your full peer review and any attached files.

**Do you want your identity to be public for this peer review?** For information about this choice, including consent withdrawal, please see our Privacy Policy.

Reviewer #1: No

Reviewer #2: **Yes: **David Restrepo
